# Can Right Heart Catheterization Improve the Prediction of Positive Response to Resynchronization Therapy?

**DOI:** 10.3390/biomedicines13020467

**Published:** 2025-02-14

**Authors:** Karolina Barańska-Pawełczak, Wojciech Jacheć, Andrzej Tomasik, Bettina Ziaja, Michalina Mazurkiewicz, Tomasz Kukulski, Celina Wojciechowska

**Affiliations:** 1Department of Cardiology, Specialistic Hospital in Zabrze, 41-800 Zabrze, Poland; bettinaziaja@gmail.com (B.Z.); michalina.liput@gmail.com (M.M.); 2Second Department of Cardiology, Faculty of Medical Sciences in Zabrze, Medical University of Silesia, 40-055 Katowice, Poland; atomasik@sum.edu.pl (A.T.); tkukulski@sum.edu.pl (T.K.); cwojciechowska@sum.edu.pl (C.W.)

**Keywords:** pulmonary hypertension, cardiac resynchronization therapy, right heart catheterization

## Abstract

**Background/Objectives**: Cardiac resynchronization therapy (CRT) is one of the interventional methods of heart failure (HF) treatment, with the criteria for CRT device implantation based on the value of the left ventricular ejection fraction, New York Heart Association functional class, QRS complex duration, and electrocardiographic morphology. Pulmonary hypertension is an important factor influencing the prognosis of patients with HF, but its influence on CRT is not fully understood. **Aim:** The main aim of the study was to determine the prognostic value of baseline right heart catheterization-derived parameters on the response to CRT. **Methods**: It was a single-centre study with retrospective analysis of data of 39 non-ischemic HF patients. Clinical, biochemical, echocardiographic, electrocardiographic, and hemodynamic data were obtained before the CRT device implantation, and after 6 months of follow-up, non-invasive re-assessment was performed. Various criteria for the response to CRT were assessed along with the correlation between the baseline parameters. **Results**: After follow-up, a significant difference was found in the reduction in symptoms associated with HF, an increase achieved in the six-minute walk test distance, and a reduction in N-terminal pro-brain natriuretic peptide concentration as well as improvement of LV function assessed in echocardiographic examination. Among all parameters assessed, the baseline higher value of the transpulmonary gradient and pulmonary vascular resistance most often had a significant negative impact on meeting the criteria of response to CRT. **Conclusions**: The results of the analyses show that the initial assessment of pulmonary hemodynamics may be crucial in predicting the response to CRT in patients with non-ischemic cardiomyopathy.

## 1. Introduction

Cardiac resynchronization therapy (CRT) is one of the interventional methods of heart failure (HF) treatment, with proven effectiveness in reducing morbidity and mortality in selected patients. According to current guidelines, the indication criteria for CRT implantation are based on the value of the left ventricular ejection fraction (LVEF), New York Heart Association (NYHA) functional class, QRS complex duration, and appropriate electrocardiographic morphology; however, the factors predisposing to a positive response to CRT are more complicated. Patients with non-ischemic HF etiology, longer QRS complex duration, and left bundle branch block (LBBB) morphology, as well as some echocardiographic patterns such as apical rocking or septal flash, have a better response to CRT [1,2]. CRT improves left ventricular (LV) function, but the importance of pulmonary circulation parameters and right ventricular (RV) function in patients with HF in the prognosis and symptoms cannot be ignored. 

Pulmonary hypertension due to left heart disease (PH-2) is a common condition associated with HF, leading to a worse prognosis regardless of the severity of LV dysfunction. The prevalence of PH-2 in patients with HF varies depending on the studied population and the severity of LV dysfunction. Parameters that have the main impact on the development of PH-2 are LV filling pressures and degree of mitral regurgitation; therefore, appropriate HF therapy may reduce pressures in the pulmonary circulation, depending on the advancement of pulmonary arteries’ remodelling [3,4]. The parameter taken into account in the assessment of pulmonary hypertension in patients with CRT most often is pulmonary arterial systolic pressure (sPAP) obtained from echocardiography. Literature data support the fact that an increase in the sPAP value at the time of CRT implantation is an unfavourable prognostic factor, and that a decrease in sPAP during CRT is a factor reducing the risk of hospitalization or death [5]. Patients with higher sPAP also have the chance to improve LVEF to a greater extent than those with a lower value; however, the sPAP value does not have a statistically significant impact on reverse LV remodelling [6,7].

As a consequence of the increased pressure in the pulmonary circulation, secondary RV dysfunction may develop over time. Available data from studies clearly suggest that patients with CRT and right ventricular dysfunction have a higher mortality than those with isolated left ventricular systolic dysfunction [8]; however, baseline assessment of RV function using echocardiographic parameters such as a tricuspid annular plane excursion (TAPSE), basal strain, fractional area change (FAC), or RV ejection fraction (RVEF) does not correlate with response to CRT and improvement of LV function [9].

It should be emphasized that there are no unified criteria for the response to CRT; various clinical or laboratory parameters can be taken into account, although reverse LV remodelling evaluation by echocardiography is used most often. The literature provides many parameters with different predictive values. One of the most commonly used parameters is LV end-systolic volume (LVESV) reduction ≥15%, although it has been shown that several percentage cut-offs of LVESV changes displayed an independent prognostic value for the prognosis. A significantly better prognostic value for reducing the risk of cardiovascular death is demonstrated by an LVEF increase of >10 U and the category of an LVEF increase of ≥1 (severe—LVEF ≤ 30, moderate—LVEF of 31–40%, mild LV dysfunction—LVEF of 41–55%, and normal LV function—LVEF ≥ 56%) [10].

Data on the effects of right heart catheterization (RHC)-derived parameters on CRT are limited. RHC in patients with left-sided HF is used mainly in qualifying patients for heart transplantation and left ventricular assistive device implantation, but it can also be useful in the assessment of valvular and congenital heart defects as well as in the differential diagnosis between restrictive cardiomyopathy and constrictive pericarditis or for myocardial biopsy. RHC allows for the diagnosis of pulmonary hypertension and assessment of disease severity, as well as determining the prognosis and response to this therapy. The question remains whether data obtained from RHC can be additionally used to find CRT responders [11,12]. Searching for new prognostic factors for patients treated with CRT is particularly important because they belong to a group at high risk of adverse cardiovascular events and RHC is a valuable tool for assessing cardiovascular function.

The aim of the study was the evaluation of usefulness of hemodynamic parameters of pulmonary circulation as predictors of positive response to CRT, assessed by commonly used criteria, in 6-month follow-up.

## 2. Study Group and Methods

It was a single-centre study with retrospective analysis of data of HF patients hospitalized in the 2nd Department of Cardiology, Faculty of Medical Sciences, in Zabrze, Medical University of Silesia, in Katowice, between 2009 and 2016.

### 2.1. Study Population

From the entire population of 155 patients suffering due to non-ischemic cardiomyopathy, with the invasive evaluation of right heart hemodynamic parameters according to qualification for heart transplantation, 39 of them met criteria for the CRT (Figure 1). 

The inclusion criteria were an age over 18 years during hospitalization; meeting the criteria required for the implantation of a CRT device including symptomatic HF defined as an NYHA functional class above II, LVEF ≤ 35%, and having the LBBB QRS morphology with QRS complex duration ≥ 130 ms; and hemodynamically stable and undergoing optimal pharmacotherapy for HF for at least three months, consisting of angiotensin-converting enzyme inhibitors (ACE-Is), angiotensin receptor blockers (ARBs), beta-blockers (BBs), and mineralocorticoid-receptor antagonists (MRAs). The exclusion criteria for the study were an age below 18 years, ischemic LV dysfunction, and lack of RHC measurements. 

The study was approved by the Ethics Committee of Silesian Medical University (NN-6501-31/05). Written, informed consent was obtained from all enrolled patients before screening. 

### 2.2. Baseline Assessment

Baseline assessment of patients included: clinical assessment included physical examination, pharmacological treatment, six-minute walk test distance (6MWT), echocardiography, electrocardiography, and RHC. The concentration of N-terminal pro-brain natriuretic peptide (NT-proBNP) was determined by the routine technique. 

#### 2.2.1. Echocardiography

Transthoracic echocardiography (TTE) images were acquired in standard views recommended by the American Society of Echocardiography and the European Association of Cardiovascular Imaging and were obtained by the use of a Vivid 7 ultrasound system (GE Healthcare Technolgies, Chicago, IL, USA) before the CRT device implantation and also within 2 days and 6 months after the procedure. The examination included LV end-diastolic diameter (LVEDD) and end-systolic diameter (LVESD) measured in the parasternal long axis view as well as LV end-diastolic volume (LVEDV) and LVESV assessed with Simpson’s biplane method of discs from two- and four-chamber views and subsequently LVEF was calculated. Pulsed-wave (PW), continuous-wave (CW), and colour Doppler imaging techniques were used to evaluate valve function as well as atrio-ventricular and interventricular dyssynchrony. The aortic velocity time integral (VTI) was measured in the five-chamber view by the pulse wave Doppler technique by tracing the VTI spectral display profile of the aorta to assess LV stroke volume. Diastolic filling time (DFT) defined as time from the beginning of the E-wave to the end of the A-wave was measured using PW-Doppler and presented as a percentage of the RR interval (DFT/RR %). Interventricular mechanical delay (IVMD) was calculated as a difference based on the pre-ejection period (PEP)—time between the beginning of electrical activation (onset of Q-wave in ECG) and the onset of LV outflow and similarly time between the beginning of QRS and the onset of RV outflow [13].

#### 2.2.2. Right Heart Catheterization

All patients underwent RHC prior to the CRT device implantation by the use of a Swan–Ganz catheter (Star Edwards Lifesciences) administered under local anesthesia (1% Lignocaine) via the right jugular vein into the pulmonary artery. After the stabilization of circulation, the following parameters were measured: right atrium pressure (RAP), systolic and diastolic RV pressures (RVs, RVd), systolic and diastolic pulmonary artery pressures (sPAP, dPAP), pulmonary artery wedge pressure (PAWP), and heart rate (HR). Cardiac output (CO) was measured by the thermodilution method using the rapid bolus injection of 10 cc of cold saline. Systolic and diastolic arterial pressure (sAP and dAP) were measured non-invasively. Hemodynamic parameters were acquired five times—mean values were used for final evaluation. Acquired data enabled the calculation of the mean pulmonary artery pressure (mPAP), mean systemic arterial pressure (mAP), transpulmonary gradient (TPG), pulmonary vascular resistance (PVR), total pulmonary resistance (TPR), systemic vascular resistance (SVR), and stroke volume index (SV) using the following formulas: mPAP = dPAP + [sPAP − dPAP]/3, mAP = dAP + [sAP − dAP]/3, TPG = mPAP − PAWP, PVR = TPG/CO, TPR = mPAP/CO, SVR = (mAP − RAP)/CO, and SV = CO/HR. Blood pressure parameters were expressed in millimetres of mercury (mmHg), CO as litres per minute (L/min), and HR as the number of heart beats per minute (bpm). Measured parameters of resistance were expressed in Wood’s units (WU) and SV in millilitres (mL) [14]. 

#### 2.2.3. CRT Device Implantation and Optimization

Device implantations were performed under local anesthesia with standard techniques. Leads (coronary sinus, right atrial, RV) were implanted via left subclavian veins. Within 24 h after CRT device implantation, routine follow-up investigations were performed, which included impedance, sensing, and threshold measurements. Atrio-ventricular delay (AVD) was adjusted to optimize LV diastolic filling, which was assessed by the use of pulsed-wave Doppler echocardiography. The goal was to maximally extend mitral inflow without the premature termination of the A-wave. Interventricular asynchrony was determined as described previously and interventricular delay (VVD) was adjusted between 0 and 80 ms to gain the minimal value and the maximal VTI. The study population included 20 patients with CRT-D and 19 patients with CRT-P. Decisions regarding the indications for the implantation of any of the devices were made individually, but the method of patient enrollment in the study and follow-up was the same.

### 2.3. Follow-Up

After six months from CRT implantation, patients underwent re-evaluation, which included clinical assessment, physical examination, electrocardiography, pharmacological treatment, 6MWT, echocardiography, and NT-proBNP concentration. In the evaluation of patients, various criteria of positive response to CRT available in the literature were taken into account: an increase in 6MWT distance of at least 10%; reduction in NT-proBNP concentration of at least 30%; reduction in the NYHA functional class of at least 1; reduction in LVEDV of at least 15%; decrease in LVESV of at least 10%, 15%, or 30%; and also assessed LVEF increase of at least 5%, 10%, or 15%. 

### 2.4. End of Study

The study was completed after the re-evaluation of patients. 

### 2.5. Statistical Analysis

The Shapiro–Wilk test showed a nonlinear distribution of continuous data; therefore, they were presented as a median with the first and third quartiles (Q1; Q3) and were compared with the Wilcoxon test. Categorical data were presented as absolute numbers and percentages and compared using the Chi-square test. Linear logistic regression analysis was used to identify variables associated with response to CRT. The results of the regression were reported as an odds ratio (OR) with corresponding 95% confidence intervals (CIs) and *p* < 0.05 was considered as statistically significant. The receiver operating characteristic (ROC) curve was determined for TPG and PVR with the calculation of the area under the curve (AUC) and determination of the intersection point of the curves for sensitivity and specificity. Data were presented with 95% CI. Spearman’s rank correlation was used for assessing the dependency between LVEF after 6 months of CRT and TPG or PVR. Statistical analysis was performed using Statistica 13.3 (TIBCO Software Inc., Krakow, Poland).

## 3. Results

The study included a group of 39 patients (8 females) with a mean age of 50.20 ± 8.20 years. All patients survived study follow-up. The patients were predominantly in the NYHA II or III functional class. The baseline (before CRT device implantation) echocardiographic, electrocardiographic, and hemodynamic data are presented in Table 1. 

After 6 months, the improvement in cardiac synchrony assessed by electrocardiographic and echocardiographic parameters resulted in a significant reduction in the LV dimensions and improvement in its contractility. A statistically significant reduction in symptoms associated with HF (assessed in NYHA classes) was found, as well as an increase in the distance achieved in the 6WMT and a reduction in NT-proBNP concentration (Table 2).

Depending on the criteria of response to therapy (responder), the size of the groups varied. The largest group was the group of patients with an increase in LVEF above a 5% absolute value; the least frequent was the increase in the 6MWT above 10% of the initial value (Table 3).

Regardless of the selected criteria for response to CRT, TPG and PVR were the variables whose initial value most often had a significant impact on meeting these criteria. Baseline data such as the NYHA functional class, NT-proBNP concentration, QRS complex duration, DFT/RR, LVEDD, and LVEF, as well as SV, PAWP, or CO, had no significant impact on the fulfilment of any of the criteria on being a responder (Table 4). Results of ROC analysis for TPG and PVR are presented in Table 5. Using Spearman’s rank correlation coefficient, a moderate negative relationship between the LVEF value 6 months after CRT implantation and the baseline PVR and TPG values was found (Figure 2 and Figure 3).

## 4. Discussion

Among the analyzed data, the vast majority of clinical, electrocardiographic, and echocardiographic parameters as well as NT-proBNP concentration showed no predictive value for the response to CRT. The distance achieved in 6MWT and some hemodynamic parameters showed an association with several criteria of the response to CRT, and among them, TPG and PVR seem to have the best predictive value. To better understand these results, it is worthwhile to analyze the changes occurring in the pulmonary circulation in patients with HF. Long-term exposure to elevated end-diastolic pressure in the LV leads to the increase in retrograde perfusion in pulmonary veins and the development of post-capillary PH (“passive” PH); in subsequent stages, to overloading of the capillaries and pulmonary arteries and the addition of a “reactive” PH component; and in the final stage of HF, to RV dysfunction [15]. TPG defined as the difference between mPAP and PCWP is the gradient between the mean pressure in the pulmonary arterial bed and the mean pressure in the left atrium. The study showed that the increase in TPG shows a negative correlation (r = −0.462; *p* = 0.004) with the improvement of LVEF after CRT device implantation, which can be explained by the fact that the increases in the “reactive” PH component reflect a bigger LV remodelling in these patients. It should be remembered that the relationship between TPG and CO may show a linear relationship in a certain range of values, but it is usually difficult to predict; that is why PVR is a much more universal and repeatable parameter, because the relationship between CO and TPG is linear, independent of the left atrial pressure, which translates into a stable PVR value [16]. In the conducted analysis, PVR showed a better negative correlation with LVEF improvement than TPG (r = −0.507; *p* = 0.001). 

The relationships between the parameters obtained in RHC and CRT should be considered in two ways: on the one hand, they have this prognostic value for the response to CRT; on the other hand, CRT implantation also has a significant impact on pulmonary circulation and RV function. Data available in the literature regarding the influence of parameters obtained during RHC on the prognosis of CRT are very limited. It was proven that patients with TPG > 12 mmHg measured within 6 months before CRT implantation have a significantly increased risk of reaching the composite endpoint (HR: 3.0; 95% CI: 1.4–6.3; *p* = 0.004) and of all-cause mortality (HR: 3.2; 95% CI: 1.3–7.4; *p* = 0.009) and also a tendency towards less improvement of the NYHA class in a 2-year follow-up period compared with patients with TPG < 12 mm Hg [17]. It was also found that there is a significant reduction in mPAP and TPG values in patients showing improvement in LV reverse remodelling or NYHA class reduction after CRT device implantation [18]. An interesting parameter combining the assessment of RV function and invasive measurements of pulmonary vascular pressures is the right ventricular to pulmonary arterial (RV-PA) coupling ratio (single-beat end-systolic elastance of RV/PA: Ees/Ea), which was considered as an independent prognostic factor, and at a baseline Ees/Ea value of ≥1 was associated with an 86% response rate to CRT [19]. Some effects of CRT may be visible immediately after device implantation, e.g., reduction in resting and stress sPAP, while other effects should be expected as chronic CRT effects (reverse remodelling) become apparent. Interestingly, long-term CRT mainly affects the RV function during exercise; its ability to cope with increased load, which can be determined by the exercise-induced increase in TAPSE (pre-CRT = 19 ± 5 mm versus post-CRT = 23 ± 7 mm; *p* = 0.003); and the TAPSE/sPAP ratio reduction (pre-CRT = 1.1 ± 0.3 mm/mmHg versus post-CRT = 0.84 ± 0.133 mm/mmHg; *p* < 0.001). Resting TAPSE after 6 months is comparable to that before CRT implantation [20]. There have also been attempts to use CRT in patients with PAH, but the results are mixed and it has never been officially recommended [21,22]. Invasive pulmonary arterial pressure (PAP) measurement may also be important in the care of patients with implanted CRT—in the subgroup analysis of the CHAMPION trial, modifications to pharmacotherapy based on the results obtained from the implanted PAP sensor reduced the rate of hospitalizations by 30% over the 18-month follow-up period compared to the group with usual HF care [23].

The study has several limitations that should be mentioned. First of all, there is the size of the group. Another limitation may be the lack of a more accurate assessment of RV function and pulmonary circulation in echocardiography.

## 5. Conclusions

CRT is an important and effective method of treating patients with LV systolic dysfunction and dyssynchrony, but in the population of implant patients according to current guidelines, not all benefit from device implantation (non-responders). Optimizing the indications for CRT and searching for new prognostic factors is crucial in this aspect. In our work, we wanted to draw attention to the RV function and pulmonary circulation, which is a hemodynamic link between the RV and LV. It is known that patients with more advanced hemodynamic abnormalities of the pulmonary circulation have an unfavourable survival prognosis. In addition, we have proven that a higher initial TPG and PVR value are also associated with a significantly worse response to CRT, and a more accurate assessment of these parameters in a larger population may be an important direction for further research. It is difficult to consider the obtained results in terms of patient qualification for CRT based only on RHC measurements. They will not replace the classical parameters and assessment of dyssynchrony in an echocardiographic examination, but may facilitate the answer to the question of who will most likely respond positively to CRT and who will require early follow-up and qualification for further invasive treatment.

## Figures and Tables

**Figure 1 biomedicines-13-00467-f001:**
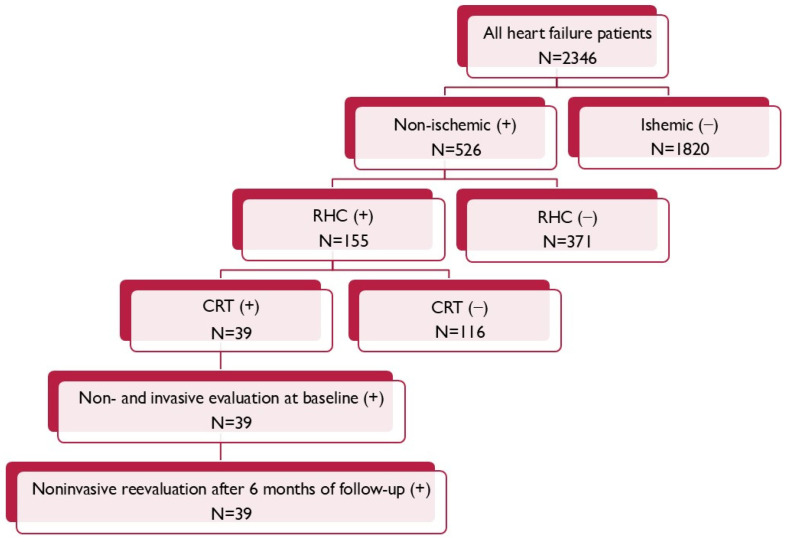
Patient flow chart.

**Figure 2 biomedicines-13-00467-f002:**
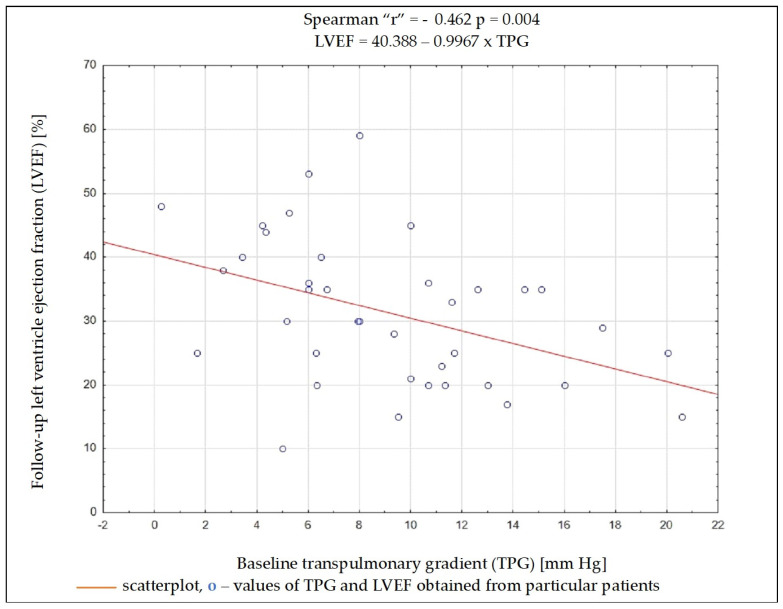
Dependency (scatterplot) between left ventricular ejection fraction after 6 months of cardiac resynchronization therapy and transpulmonary pressure gradient value before cardiac resynchronization therapy.

**Figure 3 biomedicines-13-00467-f003:**
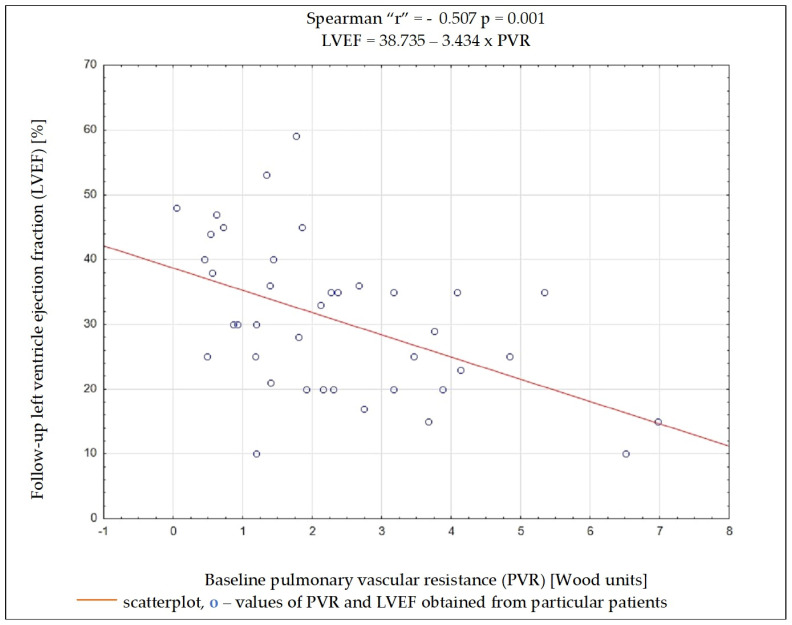
Dependency between left ventricular ejection fraction after 6 months of cardiac resynchronization therapy and pulmonary vascular resistance value before cardiac resynchronization therapy.

**Table 1 biomedicines-13-00467-t001:** Baseline characteristics of all studied groups.

Evaluated Parameters	Median (Q1–Q3) n (%)
General characteristics	
Population size	39 (100%)
Female—*n* (%)	8 (20.50%)
Age at HF onset [years]	52.46 (43.16; 55.75)
NYHA II	17 (43.59%)
NYHA III	21 (53.85%)
NYHA IV	1 (2.56%)
NT-proBNP [pg/mL]	1041 (275.0; 1863)
6MWT [m]	465.0 (409.0; 547.0)
Hypertension	14 (35.89%)
Diabetes mellitus	6 (15.38%)
Electrocardiographic and echocardiographic data	
PQ interval duration [ms]	200.0 (180.0; 220.0)
QRS complex duration [ms]	160.0 (160.0; 180.0)
DFT/RR [%]	44.00 (38.00; 47.00)
IVMD [ms]	67.00 (45.00; 75.00)
LVEDD [mm]	69.00 (64.00; 78.00)
LVESD [mm]	61.00 (55.00; 70.00)
LVEDV [mL]	230.0 (189.0; 280.0)
LVESV [mL]	170.0 (140.0; 220.0)
LVEF [%]	22.00 (17.00; 25.00)
Hemodynamic data	
sPAP [mmHg]	39.00 (32.00; 54.75)
dPAP [mmHg]	23.33 (16.00; 31.60)
mPAP [mmHg]	28.60 (20.33; 40.00)
sAP [mmHg]	125.0 (112.0; 135.0)
dAP [mmHg]	78.00 (71.67; 90.0)
mAP [mmHg]	92.00 (85.50; 103.3)
HR [min]	78.00 (70.00; 8.00)
SV [mL]	60.88 (42.47; 80.40)
PAWP [mmHg]	20.00 (14.00; 28.50)
RVs [mmHg]	40.00 (33.00; 56.00)
RVd [mmHg]	8.00 (5.00; 10.00)
PVR [WU]	1.92 (1.18; 3.46)
TPG [WU]	9.33 (6.00; 12.61)
TPR [WU]	6.20 (3.75; 10.68)
SVR [WU]	19.45 (13.97; 22.42)
RAP [mmHg]	8.00 (5.00; 12.00)
CO [L/min]	4.51 (3.40; 5.40)

6MWT—six-minute walk test distance; CO—cardiac output; dAP—diastolic arterial pressure; DFT/RR—diastolic filling time to RR-interval ratio; dPAP—diastolic pulmonary artery pressure; HF—heart failure; HR—heart rate; Q1—lower quartile; Q3—upper quartile; IVMD—interventricular mechanical delay; LVEDD—left ventricular end-diastolic diameter; LVEF—left ventricular ejection fraction; LVESD—left ventricular end-systolic diameter; LVEDV—left ventricular end-diastolic volume; LVESV—left ventricular end-systolic volume; NT-proBNP—N-terminal pro-brain natriuretic peptide; NYHA—New York Heart Association functional class; mAP—mean arterial pressure; mPAP—mean pulmonary artery pressure; PAWP—pulmonary artery wedge pressure; PQ—PQ interval; PVR—pulmonary vascular resistance; RAP—right atrium pressure; RVd—right ventricular diastolic pressure; RVs—right ventricular systolic pressure; sAP—systolic arterial pressure; sPAP—systolic pulmonary artery pressure; SV—stroke volume; SVR—systemic vascular resistance; TPG—transpulmonary pressure gradient; TPR—total pulmonary resistance; WU—Wood’s units.

**Table 2 biomedicines-13-00467-t002:** The influence of resynchronization therapy on selected functional, laboratory, electrocardiographic, and echocardiographic parameters in 6-month follow-up.

	Baseline Median (Q1–Q3) n%	Follow-Up Median (Q1–Q3) n%	Wilcoxon Test Chi-Square Test *p*
NYHA I/II	17 (43.59%)	34 (87.18%)	<0.001
NYHA III/IV	22 (56.41%)	5 (12.82%)	<0.001
6MWT [m]	465.0 (409.0–547.0)	515.5 (462.0–567.5)	<0.001
NT-proBNP [pg/mL]	1041 (275.0–1863)	587 (104–1420)	0.018
QRS complex [ms]	160.0 (160.0–180.0)	160.0 (140.0–160.0)	<0.001
PQ _(baseline)_ AVD _(follow-up)_ [ms]	200.0 (180.0–220.0)	95.00 (70.00–120.0)	<0.001
DFT/RR [%]	44.00 (38.00–47.00)	48.00 (44.00–51.00)	0.002
IVMD [ms]	67.00 (45.00–75.00)	15.00 (8.00–35.00)	<0.001
LVEDV [mL]	230.0 (189.0–280.0)	170.0 (137.0–240.0)	<0.001
LVESV [mL]	170.0 (140.0–220.0)	117.0 (80.00–170.0)	<0.001
LVEDD [mm]	69.00 (64.00–78.00)	63.00 (57.00–74.00)	0.004
LVESD [mm]	61.00 (55.00–70.00)	53.00 (46.00–67.00)	<0.001
LVEF [%]	22.00 (17.00–25.00)	30.00 (21.00–38.00)	<0.001

6MWT—six-minute walk test distance; AVD—PQ programmed by atrio-ventricular delay; DFT/RR—diastolic filling time to RR-interval ratio; Q1—lower quartile; Q3—upper quartile; IVMD—interventricular mechanical delay; LVEDD—left ventricular end-diastolic diameter; LVEF—left ventricular ejection fraction; LVESD—left ventricular end-systolic diameter; LVEDV—left ventricular end-diastolic volume; LVESV—left ventricular end-systolic volume; NT-proBNP—N-terminal pro-brain natriuretic peptide; NYHA—New York Heart Association functional class; PQ—PQ interval.

**Table 3 biomedicines-13-00467-t003:** The size of subgroups depending on various criteria of response to resynchronization therapy.

	Responder	Non-responder
6MWT ↑ ≥10%	10 (25.64%)	29 (74.36%)
NT-proBNP ↓ ≥30%	16 (41.03%)	23 (58.97)
NYHA ↓ by I	13 (33.33%)	26 (66.67%)
LVEDV↓ ≥15%	19 (48.72%)	20 (51.28%)
LVESV↓ ≥10%	23 (58.97%)	16 (41.03%)
LVESV↓ ≥15%	21 (53.85%)	18 (46.15%)
LVESV ↓ ≥ 30%	16 (41.03%)	23 (58.97%)
LVEF ↑ ≥5%	27 (69.23%)	12 (30.77%)
LVEF ↑ ≥10%	25 (64.10%)	14 (35.90%)
LVEF ↑ ≥15%	25 (64.10%)	14 (35.90%)

↑—growth; ↓—decrease; 6MWT—six-minute walk test distance; LVEDV—left ventricular end-diastolic volume; LVEF—left ventricular ejection fraction; LVESV—left ventricular end-systolic volume; NT-proBNP—N-terminal pro-brain natriuretic peptide; NYHA—New York Heart Association (NYHA) functional class.

**Table 4 biomedicines-13-00467-t004:** The chance of meeting selected response criteria to cardiac resynchronization therapy. Univariable linear regression analysis of clinical, laboratory, echocardiographic, and hemodynamic parameters.

	6MWT ↑ ≥10% OR, 95% CI, *p*	NT-proBNP ↓ ≥30% OR, 95% CI, *p*	NYHA ↓ by I OR, 95% CI, *p*	LVEDV ↓ ≥15% OR, 95% CI, *p*	LVESV ↓ ≥10% OR, 95% CI, *p*	LVESV ↓ ≥15% OR, 95% CI, *p*	LVESV ↓ ≥30% OR, 95% CI, *p*	LVEF ↑ ≥5% OR, 95% CI, *p*	LVEF ↑ ≥10% OR, 95% CI, *p*	LVEF ↑ ≥15% OR, 95% CI, *p*
Female	0.786 0.126–4.885 *p* = 0.789	0.438 0.087–2.196 *p* = 0.299	1.000 0.195–5.125 *p* = 1.000	0.218 0.037–1.306 *p* = 0.085	0.250 0.049–1.281 *p* = 0.086	0.333 0.066–1.684 *p* = 0.169	0.327 0.055–1.948 *p* = 0.204	0.857 0.166–4.438 *p* = 0.849	1.158 0.228–5.882 *p* = 0.855	1.158 0.228–5.883 *p* = 0.855
Age [years]	0.989 0.902–1.085 *p* = 0.812	0.930 0.851–1.016 *p* = 0.098	0.999 0.917–1.088 *p* = 0.973	0.981 0.904–1.064 *p* = 0.626	0.952 0.873–1.039 *p* = 0.256	0.956 0.879–1.041 *p* = 0.284	0.921 0.841–1.008 *p* = 0.064	0.942 0.854–1.038 *p* = 0.212	0.952 0.869–1.042 *p* = 0.267	0.943 0.860–1.035 *p* = 0.202
NYHA	1.342 0.294–6.136 *p* = 0.695	0.788 0.201–3.099 *p* = 0.725	1.784 0.427–7.456 *p* = 0.412	0.491 0.117–2.052 *p* = 0.313	0.326 0.074–1.446 *p* = 0.127	0.311 0.069–1.402 *p* = 0.116	0.565 0.132–2.416 *p* = 0.426	0.435 0.099–1.899 *p* = 0.252	0.251 0.053–1.193 *p* = 0.072	0.556 0.135–2.289 *p* = 0.400
6MWT [m]	0.882 0.785–0.991 *p* = 0.028	1.061 0.981–1.146 *p* = 0.124	1.057 0.977–1.143 *p* = 0.156	1.103 1.008–1.207 *p* = 0.028	1.102 1.004–1.211 *p* = 0.035	1.123 1.017–1.239 *p* = 0.018	1.094 1.003–1.193 *p* = 0.035	1.045 0.962–1.134 *p* = 0.282	1.079 0.988–1.179 *p* = 0.078	1.047 0.967–1.133 *p* = 0.243
NT-proBNP [pg/mL]	1.005 0.938–1.077 *p* = 0.890	0.962 0.903–1.025 *p* = 0.218	0.976 0.914–1.043 *p* = 0.461	0.938 0.873–1.009 *p* = 0.074	0.946 0.884–1.012 *p* = 0.093	0.951 0.890–1.017 *p* = 0.126	0.949 0.884–1.020 *p* = 0.141	0.951 0.891–1.015 *p* = 0.118	0.943 0.881–1.009 *p* = 0.078	0.974 0.917–1.035 *p* = 0.378
QRS complex [ms]	0.998 0.961–1.036 *p* = 0.907	1.002 0.970–1.036 *p* = 0.888	0.982 0.947–1.019 *p* = 0.320	1.009 0.976–1.043 *p* = 0.592	0.992 0.958–1.026 *p* = 0.623	1.006 0.973–1.040 *p* = 0.722	1.008 0.975–1.043 *p* = 0.623	1.014 0.978–1.051 *p* = 0.448	1.026 0.988–1.065 *p* = 0.165	1.016 0.981–1.053 *p* = 0.360
PQ [ms]	1.005 0.977–1.033 *p* = 0.741	0.965 0.932–0.998 *p* = 0.029	0.993 0.967–1.020 *p* = 0.590	0.980 0.952–1.008 *p* = 0.134	0.992 0.966–1.019 *p* = 0.526	0.994 0.970–1.020 *p* = 0.654	0.981 0.955–1.008 *p* = 0.152	0.980 0.950–1.011 *p* = 0.179	0.988 0.962–1.015 *p* = 0.369	0.988 0.962–1.015 *p* = 0.369
DFT/RR [%]	0.959 0.859–1.071 *p* = 0.444	0.946 0.859–1.043 *p* = 0.249	0.984 0.890–1.087 *p* = 0.738	0.998 0.915–1.090 *p* = 0.971	0.989 0.902–1.084 *p* = 0.803	0.963 0.877–1.057 *p* = 0.410	1.009 0.921–1.107 *p* = 0.838	1.003 0.910–1.104 *p* = 0.955	0.985 0.898–1.082 *p* = 0.750	0.977 0.890–1.073 *p* = 0.617
IVMD [ms]	0.993 0.962–1.026 *p* = 0.677	1.005 0.977–1.033 *p* = 0.732	0.997 0.968–1.026 *p* = 0.821	1.016 0.987–1.046 *p* = 0.270	1.037 1.001–1.074 *p* = 0.038	1.039 1.002–1.077 *p* = 0.031	1.017 0.987–1.049 *p* = 0.248	1.024 0.991–1.057 *p* = 0.136	1.031 0.997–1.066 *p* = 0.061	1.015 0.985–1.045 *p* = 0.310
LVEDD [mm]	1.012 0.938–1.092 *p* = 0.752	0.956 0.891–1.026 *p* = 0.199	0.980 0.912–1.053 *p* = 0.571	0.963 0.899–1.032 *p* = 0.273	0.942 0.875–1.014 *p* = 0.101	0.946 0.880–1.017 *p* = 0.122	0.946 0.878–1.018 *p* = 0.127	0.918 0.844–0.999 *p* = 0.039	0.893 0.816–0.978 *p* = 0.012	0.963 0.897–1.035 *p* = 0.290
LVEF [%]	1.051 0.924–1.197 *p* = 0.433	1.048 0.934–1.177 *p* = 0.409	1.146 0.993–1.322 *p* = 0.054	1.016 0.907–1.137 *p* = 0.781	1.057 0.938–1.191 *p* = 0.349	1.048 0.933–1.178 *p* = 0.411	1.088 0.961–1.231 *p* = 0.169	1.024 0.905–1.159 *p* = 0.692	1.023 0.909–1.152 *p* = 0.698	1.019 0.906–1.147 *p* = 0.742
sPAP [mmHg]	0.966 0.914–1.021 *p* = 0.207	0.956 0.910–1.004 *p* = 0.060	0.951 0.900–1.005 *p* = 0.063	0.970 0.926–1.016 *p* = 0.182	0.954 0.909–1.002 *p* = 0.053	0.952 0.906–1.000 *p* = 0.044	0.913 0.854–0.975 *p* = 0.005	0.944 0.894–0.996 *p* = 0.029	0.951 0.904–1.000 *p* = 0.044	0.962 0.916–1.009 *p* = 0.100
dPAP [mmHg]	0.936 0.851–1.028 *p* = 0.154	0.924 0.851–1.003 *p* = 0.050	0.926 0.847–1.013 *p* = 0.081	0.970 0.903–1.043 *p* = 0.399	0.955 0.886–1.028 *p* = 0.207	0.955 0.887–1.028 *p* = 0.208	0.875 0.789–0.971 *p* = 0.009	0.901 0.823–0.986 *p* = 0.019	0.901 0.825–0.984 *p* = 0.017	0.938 0.867–1.014 *p* = 0.097
mPAP [mmHg]	0.946 0.874–1.024 *p* = 0.156	0.935 0.873–1.002 *p* = 0.048	0.933 0.865–1.007 *p* = 0.067	0.968 0.910–1.030 *p* = 0.294	0.950 0.891–1.014 *p* = 0.113	0.949 0.890–1.013 *p* = 0.106	0.887 0.811–0.970 *p* = 0.007	0.917 0.850–0.989 *p* = 0.020	0.922 0.857–0.991 *p* = 0.023	0.946 0.885–1.012 *p* = 0.093
SV [mL]	1.022 0.989–1.056 *p* = 0.173	1.022 0.992–1.053 *p* = 0.136	1.010 0.981–1.040 *p* = 0.469	0.984 0.957–1.013 *p* = 0.267	1.010 0.982–1.039 *p* = 0.469	1.006 0.979–1.035 *p* = 0.641	1.021 0.991–1.052 *p* = 0.150	1.029 0.994–1.065 *p* = 0.091	1.021 0.989–1.053 *p* = 0.184	0.946 0.885–1.012 *p* = 0.093
PAWP [mmHg]	0.954 0.872–1.043 *p* = 0.284	0.959 0.888–1.035 *p* = 0.269	0.961 0.886–1.043 *p* = 0.329	1.022 0.949–1.101 *p* = 0.546	0.983 0.913–1.060 *p* = 0.648	0.991 0.921–1.066 *p* = 0.800	0.939 0.865–1.020 *p* = 0.124	0.934 0.857–1.019 *p* = 0.112	0.947 0.874–1.027 *p* = 0.176	1.016 0.986–1.047 *p* = 0.295
PVR [WU]	0.597 0.314–1.132 *p* = 0.102	0.534 0.308–0.924 *p* = 0.020	0.561 0.307–1.025 *p* = 0.052	0.689 0.437–1.088 *p* = 0.098	0.657 0.417–1.034 *p* = 0.060	0.625 0.388–1.007 *p* = 0.046	0.329 0.146–0.741 *p* = 0.006	0.537 0.320–0.902 *p* = 0.015	0.543 0.324–0.908 *p* = 0.016	0.587 0.361–0.955 *p* = 0.026
TPG [mmHg]	0.924 0.788–1.084 *p* = 0.314	0.860 0.738–1.001 *p* = 0.044	0.840 0.704–1.002 *p* = 0.046	0.769 0.632–0.936 *p* = 0.007	0.844 0.723–0.985 *p* = 0.026	0.802 0.673–0.956 *p* = 0.011	0.688 0.532–0.890 *p* = 0.003	0.830 0.706–0.976 *p* = 0.020	0.805 0.678–0.956 *p* = 0.011	0.838 0.716–0.981 *p* = 0.023
RAP [mmHg]	0.812 0.659–1.000 *p* = 0.043	0.863 0.726–1.025 *p* = 0.083	0.754 0.603–0.943 *p* = 0.010	0.817 0.676–0.986 *p* = 0.030	0.934 0.797–1.094 *p* = 0.381	0.907 0.771–1.067 *p* = 0.224	0.863 0.725–1.027 *p* = 0.087	0.939 0.795–1.110 *p* = 0.447	0.964 0.822–1.130 *p* = 0.638	0.943 0.801–1.110 *p* = 0.466
CO [L/min]	1.486 0.925–2.385 *p* = 0.090	1.520 0.946–2.445 *p* = 0.074	1.216 0.792–1.867 *p* = 0.354	0.923 0.613–1.390 *p* = 0.692	1.365 0.860–2.167 *p* = 0.172	1.360 0.864–2.143 *p* = 0.169	1.502 0.947–2.382 *p* = 0.074	1.637 0.923–2.903 *p* = 0.081	1.383 0.853–2.244 *p* = 0.174	1.515 0.894–2.567 *p* = 0.110

↑—growth; ↓—decrease; 6MWT—six-minute walk test distance; CI—confidence interval; CO—cardiac output; DFT/RR—diastolic filling time to RR-interval ratio; dPAP—diastolic pulmonary artery pressure; IVMD—interventricular mechanical delay; LVEDD—left ventricular end-diastolic diameter; LVEF—left ventricular ejection fraction; LVEDV—left ventricular end-diastolic volume; LVESV—left ventricular end-systolic volume; NT-proBNP—N-terminal pro-brain natriuretic peptide; NYHA—New York Heart Association (NYHA) functional class; mPAP—mean pulmonary artery pressure; OR—odds ratio; PAWP—pulmonary artery wedge pressure; PVR—pulmonary vascular resistance; RAP—right atrium pressure; sPAP—systolic pulmonary artery pressure; SV—stroke volume; TPG—transpulmonary pressure gradient; WU—Wood’s units.

**Table 5 biomedicines-13-00467-t005:** Data from receiver operating characteristic curve analysis results for pulmonary vascular resistance and transpulmonary pressure gradient.

	Cut-Off Value	Specificity/Sensitivity [%]	AUC	95% CI
PVR [WU]				
6MWT ↑ ≥10%	1.42	70.0	0.703 ± 0.098	(0.511–0.896)
NT-proBNP ↓ ≥30%	1.80	68.4	0.734 ± 0.08	(0.577–0.892)
NYHA ↓ by I	1.81	61.5	0.683 ± 0.086	(0.514–0.853)
LVEDV↓ ≥15%	1.81	66.7	0.692 ± 0.088	(0.520–0.864)
LVESV↓ ≥10%	2.12	69.2	0.709 ± 0.086	(0.541–0.877)
LVESV↓ ≥15%	1.92	71.8	0.73 ± 0.082	(0.569–0.891)
LVESV ↓ ≥ 30%	1.80	75.0	0.823 ± 0.069	(0.688–0.959)
LVEF ↑ ≥5%	2.16	66.7	0.765 ± 0.081	(0.606–0.925)
LVEF ↑ ≥10%	2.16	68.0	0.757 ± 0.081	(0.599–0.915)
LVEF ↑ ≥15%	2.12	64.2	0.737 ± 0.08	(0.580–0.894)
TPG [mmHg]				
6MWT ↑ ≥10%	7.93	61.0	0.619 ± 0.11	(0.404–0.834)
NT-proBNP ↓ ≥30%	8.0	69.2	0.717 ± 0.084	(0.552–0.882)
NYHA ↓ by I	7.93	69.2	0.701 ± 0.086	(0.533–0.869)
LVEDV↓ ≥15%	8.0	79.5	0.821 ± 0.074	(0.676–0.966)
LVESV↓ ≥10%	9.5	79.7	0.749 ± 0.086	(0.58–0.918)
LVESV↓ ≥15%	9.33	77.0	0.784 ± 0.08	(0.628–0.941)
LVESV ↓ ≥ 30%	7.93	81.9	0.856 ± 0.065	(0.729–0.983)
LVEF ↑ ≥5%	10.0	74.5	0.765 ± 0.083	(0.603–0.928)
LVEF ↑ ≥10%	9.8	72.0	0.783 ± 0.077	(0.632–0.933)
LVEF ↑ ≥15%	9.78	70.0	0.753 ± 0.08	(0.597–0.909)

↑—growth; ↓—decrease; 6MWT—six-minute walk test distance; AUC—area under the curve; CI—confidence interval; LVEF—left ventricular ejection fraction; LVEDV—left ventricular end-diastolic volume; LVESV—left ventricular end-systolic volume; NT-proBNP—N-terminal pro-brain natriuretic peptide; NYHA—New York Heart Association (NYHA) functional class; PVR—pulmonary vascular resistance; TPG—transpulmonary pressure gradient; WU—Wood’s units.

## Data Availability

The original contributions presented in this study are included in the article. Further inquiries can be directed to the corresponding authors.

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
