# Peer review of "Can Right Heart Catheterization Improve the Prediction of Positive Response to Resynchronization Therapy?"

_biomedicines, 2025, doi:10.3390/biomedicines13020467_

Round 1
Reviewer 1 Report
Comments and Suggestions for Authors
I have reviewed the following manuscript and suggest some recommendations
- Please improve the title more comprehensive and attractive. Does the title accurately reflect the study's scope and findings? the abstract should be concise and comprehensive?
- Please mention the objective of your study in abstract and introduction? Are the objectives of the study clearly stated in abstract and introduction?
- Why you have selected this topic. Where is your rational study. Please mention rationalization of your study in introduction.
- Please add some some data in introduction from literature and the refereces are not sufficient. Is the introduction well-structured and does it provide sufficient context for the study? Please cite some recent references.
- Is the study design appropriate to address the research question? Are the inclusion and exclusion criteria for the study population clearly defined?
- Please mentioned and cite the methods for collecting clinical, biochemical, and echocardiographic data clearly described?
- Please cite and defined the standard protocol that you have followed for this study. Was the right heart catheterization performed according to standard protocols?
- Please check once the calculation of the whole data. Are the hemodynamic parameters calculated using valid and reproducible formulas?
- Please mention the statistical parameter that you have followed for this study. Was the sample size sufficient to achieve statistical significance?
- Are the baseline characteristics of the study population comprehensively presented? Are the CRT device implantation and optimization protocols (Lines 154–163) adequately detailed and justified?
- Is the statistical analysis appropriate and well-explained? And please the tables and figures should be clearly and appropriately labeled?
- Please explain the abbreviation used in this study be clearly define once. Are the acronyms and abbreviations adequately defined?
- Are the findings of the six-minute walk test (6MWT) clearly presented? Is the reduction in N-terminal pro-brain natriuretic peptide (NT-proBNP) concentrations adequately discussed?
- Are the echocardiographic parameters and measurements (Lines 116–133) clearly explained and consistent with standard guidelines? Are the echocardiographic improvements after CRT implantation supported by robust data?
- Please mentioned the differences in hemodynamic parameters before and after CRT clearly outlined.
- Were the statistical methods for testing hypotheses appropriate? Are the p-values and confidence intervals accurately reported? Is the use of logistic regression models justified?
- Are the results of the receiver operating characteristic (ROC) curve analysis appropriately interpreted?
- Does the discussion adequately contextualize the study findings within existing literature? Please compare your results with literature and cited the references.
- Are the implications of the results for CRT therapy sufficiently explored? Please mention the article identify the limitations of the study?
- Please mentioned your novelity/ findings in your conclusion. Are the conclusions drawn from the data justified?
- Is the role of pulmonary hypertension as a prognostic factor for CRT well addressed? Are the effects of CRT on left ventricular ejection fraction (LVEF) sufficiently detailed?
- Does the study evaluate the impact of CRT on right ventricular function? Are the responder criteria for CRT adequately defined? Please add the study protocol in materials and methods.
- Please check the values that the prognostic values of TPG and PVR convincingly demonstrated? Is the relationship between baseline TPG and CRT response statistically and clinically meaningful?
- Does the study effectively use Spearman’s correlation (Lines 237–238) to assess relationships between parameters and outcomes?
- Please mention your echocardiographic techniques used for assessing dyssynchrony sufficiently described? And describe their protocol in materials and methods section
- What is the future prospects of your study?
- How can you explain that your findings are scientifically sound and beneficial to community. Compare your results with literature and answer in modern perspective.
- Please remove the typographical and grammatical errors from the paper?
- Conclusion should be rewritten and should make attractive for the reader.

should be improved
Author Response
- Please improve the title more comprehensive and attractive. Does the title accurately reflect the study's scope and findings? the abstract should be concise and comprehensive?
The title of the manuscript has been changed
Can right heart catheterization improve the prediction of positive response to resynchronization therapy?
2.Please mention the objective of your study in abstract and introduction? Are the objectives of the study clearly stated in abstract and introduction?
We believe that the purpose of the work has been clearly defined
3.Why you have selected this topic. Where is your rational study. Please mention rationalization of your study in introduction.
The introduction has been re-edited.
4.Please add some some data in introduction from literature and the refereces are not sufficient. Is the introduction well-structured and does it provide sufficient context for the study? Please cite some recent references.
The introduction has been re-edited.
5.Is the study design appropriate to address the research question? Are the inclusion and exclusion criteria for the study population clearly defined?
Yes, the longitudinal study design is appropriate to answer the research question. Although we reviewed the retrospective patient data (2009-2016) we have followed them prospectively in our out-patient department. For this analysis, we present the results of 6 months of follow-up with the patients. The inclusion and exclusion criteria are clearly defined.
6. Please mentioned and cite the methods for collecting clinical, biochemical, and echocardiographic data clearly described?
The methods for collecting clinical and echocardiographic data are clearly described. We have written that NTproBNP was measured routinely, e.g., in a certified analytical laboratory, and the baseline and follow-up values were picked from the electronic patient records.
7. Please cite and defined the standard protocol that you have followed for this study. Was the right heart catheterization performed according to standard protocols?
The study protocol is presented in Figure 1. Right heart catheterization was performed according to generally accepted guidelines.
References has been supplemented: "Zakliczynski M, Zebik T, Maruszewski M, Swierad M, Zembala M. Usefulness of pulmonary hypertension reversibility test with sodium nitroprusside in stratification of early death risk after orthotopic heart transplantation. Transplant Proc. 2005 Mar;37(2): 1346-8. doi: 10.1016/j.transproceed.2005.01.012. PMID: 15848716.
8. Please check once the calculation of the whole data. Are the hemodynamic parameters calculated using valid and reproducible formulas?
The correctness of the calculations was checked.
Hemodynamic parameters were calculated using standard, commonly used formulas
9. Please mention the statistical parameter that you have followed for this study. Was the sample size sufficient to achieve statistical significance ?
We have analyzed many variables to follow longitudinally in our research. They are listed in Table 2. The sample size is limited, and we have acknowledged this in the Discussion section, though we have used non-parametric tests to analyze our data and assess as reliable as possible conclusions (Statistical Methods, Snedecor and Cochrane, 7th ed., The Iowa State University Press, 1980)
11. Are the baseline characteristics of the study population comprehensively presented? Are the CRT device implantation and optimization protocols (Lines 154–163) adequately detailed and justified?
We have characterized our study group extensively in text and table, as well as CRT implantation and its optimization protocols.
12. Is the statistical analysis appropriate and well-explained? And please the tables and figures should be clearly and appropriately labeled?
All statistical computations were performed using Statistica 13.3 software from TIBCO, licensed to the Medical University of Silesia. Yet, the basic statistical reasoning and principles were from “Statistical Methods” by Snedecor and Cochrane (cited above), and “PrzystÄ™pny kurs statystyki” by Stanisz, Statsoft Polska Sp. z o. o. Kraków 2007.
13. Please explain the abbreviation used in this study be clearly define once. Are the acronyms and abbreviations adequately defined?
Abbreviations used in the text have been unified and corrected.
15. Are the findings of the six-minute walk test (6MWT) clearly presented? Is the reduction in N-terminal pro-brain natriuretic peptide (NT-proBNP) concentrations adequately discussed?
6MWT was performed according to generally accepted protocol
It has been already mentioned , that baseline 6MWT was associated with some parameters of CRT response , in particular volumetric response (LVEDV >15%, LVESV >10,15,30%) ,see results in the table 4
16. Are the echocardiographic parameters and measurements (Lines 116–133) clearly explained and consistent with standard guidelines? Are the echocardiographic improvements after CRT implantation supported by robust data?
Echo measurements are consistent with current guidelines. Additionally ,LVOT measurement was performed in LAX parasternal view ,3 mm below the aortic annulus
I hereby confirm that echo measurements performed before and after CRT are supported by robust data
17. Please mentioned the differences in hemodynamic parameters before and after CRT clearly outlined.
The study protocol (Figure 1) did not include performing right heart catheterization after CRT implantation. (Study group and Methods chapter ; paragraphs 2.2 and 2.4). We do not have these data.
18. Were the statistical methods for testing hypotheses appropriate? Are the p-values and confidence intervals accurately reported? Is the use of logistic regression models justified?
All statistical computations were performed using Statistica 13.3 software from TIBCO, licensed to the Medical University of Silesia. Yet, the basic statistical reasoning and principles were from “Statistical Methods” by Snedecor and Cochrane (cited above), and “PrzystÄ™pny kurs statystyki” by Stanisz, Statsoft Polska Sp. z o. o. Kraków 2007.
19. Are the results of the receiver operating characteristic (ROC) curve analysis appropriately interpreted?
All statistical computations were performed using Statistica 13.3 software from TIBCO, licensed to the Medical University of Silesia. Yet, the basic statistical reasoning and principles were from “Statistical Methods” by Snedecor and Cochrane (cited above), and “PrzystÄ™pny kurs statystyki” by Stanisz, Statsoft Polska Sp. z o. o. Kraków 2007.
20. Does the discussion adequately contextualize the study findings within existing literature? Please compare your results with literature and cited the references.
Data on the correlation between pulmonary circulation and response to CRT in the vast majority concern parameters obtained in echocardiographic examination. There are only few studies on the assessment of the relationship between invasive measurements of pulmonary circulation and CRT, all of these studies as well as case reports have been thoroughly analyzed.
21. Are the implications of the results for CRT therapy sufficiently explored? Please mention the article identify the limitations of the study?
The results may allow for the selection of patients with particularly poor prognosis for CRT. Limited studies are presented in the text.
22. Please mentioned your novelity/ findings in your conclusion. Are the conclusions drawn from the data justified?
We believe that already at the stage of CRT device implantation we could be able to select HF patients with a particularly poor prognosis and consider additional therapeutic options for them (qualification for LVAD or heart transplant).
23. Is the role of pulmonary hypertension as a prognostic factor for CRT well addressed? Are the effects of CRT on left ventricular ejection fraction (LVEF) sufficiently detailed?
The study focused not on assessing the effect of pulmonary hypertension on CRT but on the role of parameters obtained from invasive measurement of pressures in the pulmonary circulation. The actual values ​​of pressures in the pulmonary circulation appeared to have less influence on achieving specific response points to CRT than PVR and TPG.
Are the effects of CRT on left ventricular ejection fraction (LVEF) sufficiently detailed? – YES.
24. Does the study evaluate the impact of CRT on right ventricular function? Are the responder criteria for CRT adequately defined? Please add the study protocol in materials and methods.
The aim of the study was not to assess the effect of CRT on right ventricular function, especially since echocardiography would be necessary for this, and this was not the aim of the work. We wanted to focus on the role of pulmonary circulation parameters obtained in invasive measurements.
25. Please check the values that the prognostic values of TPG and PVR convincingly demonstrated? Is the relationship between baseline TPG and CRT response statistically and clinically meaningful?
The results we obtained for 39 patients were analyzed univariably, so we have not concluded in the manuscript that TPG and PVR remain independent predictors of CRT therapy response. We have written more liberal sentences: “It is difficult to consider the obtained results in the aspect of qualifying patients for CRT based on the measurements obtained from RHC; they will not replace the classic assessment of dyssynchrony in an echocardiographic examination, but they may facilitate the answer to the question who will respond to the therapy.”
A larger study population would allow more statistical power.
26. Does the study effectively use Spearman’s correlation (Lines 237–238) to assess relationships between parameters and outcomes?
The Spearman’s correlation seemed the most appropriate for presenting parameters and the outcomes. However, this is only for illustrative purposes.
27. Please mention your echocardiographic techniques used for assessing dyssynchrony sufficiently described? And describe their protocol in materials and methods section
The echocardiographic study protocol together with the parameters assessed are presented in the text. The detailed assumptions regarding the measurement of each echocardiographic parameter are, in our opinion, beyond the scope of this manuscript, but a footnote to the ASE guidelines on which they are based is included in the text: Gorcsan J 3rd, Abraham T, Agler DA, Bax JJ, Derumeaux G, Grimm RA, Martin R, Steinberg JS, Sutton MS, Yu CM; American Society of Echocardiography Dyssynchrony Writing Group. Echocardiography for cardiac resynchronization therapy: recommendations for performance and reporting--a report from the American Society of Echocardiography Dyssynchrony Writing Group endorsed by the Heart Rhythm Society. J Am Soc Echocardiogr. 2008 Mar;21(3):191-213. doi: 10.1016/j.echo.2008.01.003. PMID: 18314047.
28. What is the future prospects of your study?
CTR is a therapeutic method with proven effectiveness in patients with HF. Thanks to the results obtained from RHC, we believe that already at the stage of CRT device implantation we could be able to select HF patients with a particularly poor prognosis and consider additional therapeutic options for them (qualification for LVAD or heart transplant).
29. How can you explain that your findings are scientifically sound and beneficial to community. Compare your results with literature and answer in modern perspective.
Thanks to the results obtained from RHC, we believe that already at the stage of CRT device implantation we could be able to select HF patients with a particularly poor prognosis and consider additional therapeutic options for them (qualification for LVAD or heart transplant).
30. Please remove the typographical and grammatical errors from the paper?
The work has been checked
31. Conclusion should be rewritten and should make attractive for the reader.
The conclusion has been re-edited.
Reviewer 2 Report
Comments and Suggestions for Authors
Interesting study with some novelity
The sample is very small- 39 patients
I do not think that every patients with advanced HF needs RT heart catetherisation. The procedure is invasive and some times can be complications.
There are other criteria for CRT or CRTD (there vere no patients with CRTD ??)
like QRS widht and others
Can you discuss this points please
Author Response
I do not think that every patients with advanced HF needs RT heart catetherisation. The procedure is invasive and some times can be complications.
We agree with the statement that not every patient with advanced HF requires RHC, especially since it is an invasive procedure. As shown in the flow chart, out of 526 patients with non-chemic HF, only 155 ultimately underwent RHC, and only because it was one of the elements of the evaluation for potential heart transplantation, and not a general diagnostic test.
There are other criteria for CRT or CRTD (there vere no patients with CRTD ??) like QRS widht and others. Can you discuss this points please
The study population included patients with CRT-D (20 patients with average LVEF 22%) and CRT-P (19 patients with average LVEF 22%). Decisions regarding the indications for implantation of any of the devices were made individually and based on the presumed risk for sudden cardiac death, but the method of patient enrollment in the study and follow-up was the same.
Round 2
Reviewer 1 Report
Comments and Suggestions for Authors
Accept in present form.
Comments on the Quality of English Languageimproved
Reviewer 2 Report
Comments and Suggestions for Authors
acceptable